behaviour/ecology/evolution

between-group competition, territoriality, chimpanzees, bisexually-bonded social system

**Authors for correspondence:**
Sylvain Lemoine
e-mail: sylvain_lemoine@eva.mpg.de
Roman M. Wittig
e-mail: wittig@eva.mpg.de

†Lead contact.

## Group dominance increases territory size and reduces neighbour pressure in wild chimpanzees

Sylvain Lemoine[1,2,3,†], Christophe Boesch[1], Anna Preis[1,3], Liran Samuni[1,3], Catherine Crockford[1,2,3] and Roman M. Wittig[1,2,3]

[1]Department of Primatology, and [2]Department of Human Behavior, Ecology and Culture, Max Planck Institute for Evolutionary Anthropology, Deutscher Platz 6, 04103 Leipzig, Germany
[3]Taï Chimpanzee Project, Centre Suisse de Recherche Scientifique en Côte d'Ivoire, 01 BP 1303, Yopougon, Abidjan, Ivory Coast

SL, 0000-0001-9853-5246; CC, 0000-0001-6597-5106; RMW, 0000-0001-6490-4031

Territorial social species, including humans, compete between groups over key resources. This between-group competition has evolutionary implications on adaptations like in-group cooperation even with non-kin. An emergent property of between-group competition is group dominance. Mechanisms of group dominance in wild animal populations are difficult to study, as they require long-term data on several groups within a population. Here, using long-term data on four neighbouring groups of wild western chimpanzees, we test the hypothesis that group dominance impacts the costs and benefits of between-group competition, measured by territory size and the pressure exerted by neighbouring groups. Larger groups had larger territories and suffered less neighbour pressure compared with smaller groups. Within-group increase in the number of males led to territory increase, suggesting the role of males in territory acquisition. However, variation in territory sizes and neighbour pressure was better explained by group size. This suggests that the bisexually-bonded social system of western chimpanzees, where females participate in territorial behaviour, confers a competitive advantage to larger groups and that group dominance acts through group size in this population. Considering variation in social systems offers new insights on how group dominance acts in territorial species and its evolutionary implications on within-group cooperation.

# 1. Introduction

Many social species compete for resources between groups of conspecifics [1–3]. This between-group competition (BGC) has evolutionary implications, especially for humans, resulting in forms of territoriality and probably shaping aspects of sociality, such as within-group cooperative skills [4,5]. Consequently, understanding the mechanisms and effects of BGC in social species is of key importance but requires long-term studies on several groups within a population [6]. The inter-group dominance hypothesis [7] is a theoretical framework attempting to integrate BGC into an evolutionary canvas and proposes that neighbouring groups compete for space and that competition results in a population-scale group hierarchy. Dominant groups could then benefit from larger territories and therefore increased access to food resources. This would regulate within-group feeding competition, which in turn would benefit individuals through nutritional benefits causing improved fitness [7,8]. Other benefits, such as mate-attraction and infant-defence [9], might also be associated with a high group dominance status.

A variety of social species show evidence of group dominance effects: (i) African lionesses (*Panthera leo*) that live in larger prides face lower levels of BGC, have access to better quality habitats and show lower mortality [8]; (ii) large group sizes in spotted hyenas (*Crocuta crocuta*) have positive effects on hunting success and reproduction [10]; and (iii) larger packs of wolves (*Canis lupus*) are more successful than smaller packs in preying upon large prey [11] and larger packs outcompete smaller packs of neighbouring conspecifics [12], potentially resulting in fitness benefits. Among primates, effects of group dominance through large group sizes have also been found: (i) larger groups of vervet monkeys (*Chlorocebus pygerythrus*) inhabit higher quality habitats and have lower infant and juvenile mortality [13]; (ii) larger groups of Japanese macaques (*Macaca fuscata*) occupy better quality habitats which entail lower travelling costs associated with BGC [14]; (iii) larger groups of black and white colobus (*Colobus guereza*) seem dominant and access better quality feeding areas [15]; and (iv) larger groups of wedge-capped capuchins (*Cebus olivaceus*) benefit from improved reproductive success than smaller groups [16].

Chimpanzees (*Pan troglodytes*) are one of the most territorial primate species, with intense BGC [17], which can lead to inter-group killings [18–21]. Chimpanzees live in a fission–fusion system, where individuals roam within their territory in parties of fluctuating sizes and composition [22,23]. This system can create strong, but temporary, power imbalances between neighbouring groups, resulting in low between-group aggression costs from the outnumbering parties in cases of strong imbalance [24]. In chimpanzees, adult males seem to play an important role in group dominance: inter-group aggression is more likely to occur when the number of males in a group is high [21]; in Ngogo (Uganda), inter-group killings by males seemed specifically targeted toward a small neighbouring group, leading across time to a territorial expansion [25], suggesting that the Ngogo group dominated the smaller one. In another eastern chimpanzee (*Pan t. schweinfurthii*) population (Kanyawara, Uganda), play-back experiments simulating inter-group encounters also suggested the preponderant role of males in responding to intrusion of neighbours [26]. Even if territorial expansion in Ngogo chimpanzees seems caused by the number of males [25], the relationship between territory size and demographic competitive ability in chimpanzees remains unclear, with inconsistent evidence across populations. In Gombe eastern chimpanzees (Tanzania), no relationship between the number of males and territory size was detected [27], but instead the number of females was positively associated with territory size. In the North group of western chimpanzees (*Pan t. verus*) from the Taï National Park (Côte d'Ivoire), while the number of adult females was negatively related to territory size, an increase in the number of adult males was more meaningful in explaining territory size increase [23,28].

Although inter-group killings are more frequent in eastern chimpanzees [21], they also occur in western chimpanzees [20]. These differences may come from population density differences [21], larger numbers of adult males in eastern populations [21], but also from differences in grouping patterns [29]. Chimpanzee populations differ in their social grouping patterns (summarized in [30]): most eastern chimpanzee populations are characterized by a male-bonded community system [27,31,32], where males have a much larger territory than females. In contrast, some western chimpanzee populations, including the different groups of the Taï population [20,30,33] and the Bossou group [34], are characterized by a bisexually-bonded system, where both males and females occupy a similar territory and actively participate in territorial behaviour [20,35,36]. These differences in social structures may imply different mechanisms of group dominance. While adult males in male-bonded populations play an essential role in territorial behaviour and therefore in territorial expansion and neighbours' repulsion, in bisexually-bonded populations, group size may matter more than the number of males for the group's competitive ability. Given the importance of inter-group interactions

and BGC in human evolution [37–40], understanding the mechanisms of group dominance in our closest living relatives and how their social structures relate to their competitive ability would shed light on the mechanisms by which BGC acted as a selective pressure on territoriality and cooperation with non-kin in the hominoid lineage.

In order to test the mechanisms of group dominance, extensive long-term data on several groups of differing demography are necessary. The Taï chimpanzees offer the opportunity to test effects of group dominance in this highly territorial species. The Taï Chimpanzee Project's long-term database provides more than 50 group years of inter-group encounter observations on four neighbouring groups, with noticeable inter- and intra-group demographic variation over time [23,41]. Using these long-term data, we tested the hypothesis whether or not group dominance effects are reflected in the costs and benefits of BGC measured using territory size and perceived neighbour pressure (measured by a composite index considering the rate of inter-group encounters (IGEs), the degree of neighbour intrusion of these IGEs, and the salience of the location of IGEs in term of past usage [42]). Larger territory sizes and lower neighbour pressure are considered beneficial, while smaller territories and higher neighbour pressure are considered costly for the individuals [43,44]. As potential proxies for group dominance, determined by the competitive ability of a group, we used the number of adult individuals of the dominant sex of the group (males), the number of adults and adolescent males and females (number of mature individuals) and group size (number of independent individuals). Although the number of males is a common measure of the resource holding potential for species with male philopatry [45], in the bisexually-bonded social system of Taï chimpanzees, females actively participate in territorial maintenance [20,35,36] and can therefore contribute to a group's competitive ability. Thus, alternatively, we expect the number of mature individuals and/or group size to impact on territory size and perceived neighbour pressure.

First, we tested how annual territory sizes are influenced by a group's competitive ability (number of males, number of mature individuals and group size separately) and by food availability, two variables known to determine territory sizes in a variety of species [43]. Second, we analysed how perceived neighbour pressure, which reflects potential threat of intrusion by neighbours into one's territory [42], is influenced by a group's competitive ability, food availability and presence of attractive females. The rationale behind this approach is that BGC in chimpanzees is potentially driven by (i) the males' need to attract more fertile females, and (ii) by the group's need to ensure safe feeding grounds to support their energetic requirements. More dominant groups should be better at competing for these resources than less dominant ones, but variation in resource availability may change the intensity of competition between the groups. Despite the availability of several neighbouring groups, we could not include the relative power of each group in relation to the response variables, since all groups are also neighboured by unhabituated groups from which no information is available regarding their competitive ability. However, we modelled in our analysis potential patterns of group dominance via group differences, by disentangling the demographic within- and between-group effects [46].

# 2. Methods

## 2.1. Study site and population

We used data collected in the Taï Chimpanzee Project (TCP) [23,41], located in the Taï National Park, Côte d'Ivoire (5°45′ N, 7°7′ W), from January 1997 to October 2016 on four habituated communities of western chimpanzees (*P. t. verus*): North group (1997–2016), South group (1999–2016), Middle group (1999–2004) and East group (2008–2016)—electronic supplementary material, table S1. We are not looking specifically at interactions between these groups, as each of these groups have more neighbouring non-habituated groups, even if some of the inter-group encounters considered in this study involve these groups. The neighbour pressure index (NPI) [42] enables captures of the level of threat represented by the intrusion of any neighbours, without having the information about their number or relative power (see below).

## 2.2. Data collection

From 1997 to 2016, we used nest-to-nest continuous focal follows on individual chimpanzees [47], to record information on behaviour, party compositions, vocalizations, activity and social interactions involving the focal individual [41]. We recorded demographic changes (births, deaths, disappearance, immigration and emigration) of all group members, and group-level events, such as inter-group

and November 2013, locations of the focal individuals were reported as the GPS coordinates of the centre of a $500 \times 500$ m grid-cell system covering the entire study area [48]. From December 2013 onwards, spatial data were recorded every minute using GPS devices (Garmin® 62 or Rino). We individually checked the GPS tracks to match them with the daily observation periods.

## 2.3. Observation time

We defined observation time as the time human observers followed individual chimpanzees of different parties, keeping only one set of the data whenever different focal individuals were simultaneously followed within the same party. The observation time was calculated in hours and log-transformed, reflecting the cumulative effect of observation on ranging data in particular, expected to reach a plateau after a certain amount of observation time. During the study period (1997–2016), a total of 13 205 individual focal follow-days were sampled (North: 1115 males, 2743 females; Middle: 828 males, 602 females; South: 2044 males, 3108 females; East: 1724 males, 1041 females), with a median of 93 (IQR ± 87.25) and 146 (IQR ± 111.25) follow-days per year for males and females respectively, resulting in a median of 858.4 (IQR ± 870.8) and 1144.9 (IQR ± 994.5) hours of direct observation per year for males and females respectively, all groups included.

## 2.4. Yearly territory sizes

We chose yearly intervals (January 1 to December 31) as the temporal scale to calculate territory size, as a full year enables inclusion of the majority of food types consumed by chimpanzees, and takes into account intra-annual productivity differences [49,50]. Territories were defined as the 95% fixed-kernel of all locations visited by group members during focal follows in a given year, a comparable measure to previous studies on chimpanzee territory size [48,51]. We processed spatial datasets in R (v. 3.3.2; R Core Team) using the following packages: GISTools_0.7-4, splancs_2.01-40, adehabitatHR_0.4.14, rgdal_1.2-13. We used the R function 'kernelUD' of the package adehabitatHR (v. 0.4.14), with a smoothing factor (h) using the plug-in method [52], to determine utilization distributions. Areas were assessed with self-made functions. Borders were drawn around kernels every 10% on yearly home-ranges, by the function 'getverticehr' of the adehabitatHR package. A summary of spatial parameters is provided in electronic supplementary material, table S1.

## 2.5. Demographic parameters

We considered demographic parameters to determine the competitive ability of a given group. Taï male chimpanzees enter the social hierarchy and reach social adulthood at about the age of 12 [53], an age at which they can already reproduce and engage in territorial behaviour [23,54]. Therefore, in concordance with other studies [35,36,55–57] we considered here males as adult when the have reached 12 years of age. We considered females as adult as soon as they presented exaggerated sexual swellings (minimum age: 9.5 years old for one individual; mean age: 11 years old). Adolescent males and females included those aged between 10 and 12 years old, excluding females already presenting exaggerated sexual swellings [23]. Group size was defined as the total number of within-group weaned individuals, that is all individuals that travel and feed independently from their mothers. Numbers of males, number of mature individuals (adult and adolescent males and females) and group sizes were measured monthly and averaged across the year. All groups experienced demographic variation due to deaths, births, female emigrations and immigrations (electronic supplementary material, table S1).

## 2.6. Female attractiveness

We included two measures of female attractiveness: a 'full tumescent swelling ratio' that takes into account the variation within a group of the number of females that presented exaggerated sexual swellings; and the number of nulliparous females within a group related to group size. The first measure considers that, even if pluriparous females are unlikely to emigrate under the pressure of neighbours, they may nonetheless constitute a factor of attraction, as extra-group forced copulations have been observed, even if rare [20]. The second measure considers that neighbouring groups may be attracted by females that could potentially emigrate to their group.

To calculate the variable 'full tumescent swelling ratio', we considered females from the appearance of their first genital swellings (mean ± s.d. age: 11 ± 1 yr; min. age: 9.5 yr for one individual) and computed the ratio as the following: for each month, among all females that had already developed tumescent swellings (considered as adults), we counted how many of them could have fully tumescent swellings and how many females with fully tumescent swellings were actually observed. However, reporting just the ratio of these two values would not account for the fact that several females could have swellings the same day, so we adjusted the number of females presenting fully tumescent swellings to the number of days where a certain number of females had swellings. For example, if 18 females could have presented swellings in a given month, and if out of 30 observation days, two swelling females were observed 12 days, one swelling female was observed 7 days and zero swelling females were observed 11 days, the index becomes $\{[(11 \times 0)/18] + [(7 \times 1)/18] + [(12 \times 2)/18]\}/30 = 0.020$.

This conservative measure of the number of fully tumescent females allows circumventing the issue of not observing females for some days and so not being able to assess the complete fully tumescent period.

Nulliparous females are those from the age of 10 years old (including the one female already showing swellings at age 9.5) until they emigrate, or until they give birth to their first infant when they did not emigrate (electronic supplementary material, table S1).

## 2.7. Neighbour pressure index

We used the NPI proposed in Lemoine *et al*. [42], which reflects the potential threat due to intrusion by neighbours into one's territory. The index was calculated on all types of IGE (physical: $n = 103$; and vocal: $n = 281$, across all groups). Physical IGE included those where individuals from opposing groups are in close visual contact, with or without direct physical aggression. Vocal IGE include those where individuals from opposing groups were not in visual contact. We distinguished vocalizations and drums emitted by neighbours from those emitted by the same group's members, either because the entire group was present with the observer, or based on the direction of the emitted vocalizations relative to the group's location, and the vocal response and reaction of fear and excitement displayed by the followed group. We know from previous studies that Taï chimpanzees produce group-specific calls [58] and that they can differentiate familiar from unfamiliar calls [59]. Furthermore, Taï chimpanzees' vocal and gestural responses to neighbours differ from their response to in-group members [59].

The NPI, developed in [42], was calculated as follows: $NPI = \mu [(I) \times (K)] j \times Fj$, where 'I' represents the relative distance of the IGE to the territory centre. 'I' is measured as the distance between the location of the encounter and the centre of the territory, relative to the distance to the border delimited by the kernel distribution of 75% of the locations (used as a cut-off point to delimit core areas [28,48]). The degree of intrusion 'I' is thus independent of the size of the territories, being a relative measure, and provides a standardized measure for variable territory sizes. An 'I' of value 1 corresponds to encounters taking place at the core area border delimited by 75% of the locations. Encounters taking place inside the core area have a value >1, while encounters taking place in the 'periphery' have a value below 1. 'K' represents the past usage of the location of the IGE, expressed as the kernel in which the IGE took place. To obtain 'K', we computed utilization distribution of all the locations for the last 12 months prior to each IGE and used the inverse of the kernel in which the IGE occurs (as lower kernel values correspond to more heavily used areas). The usage of the territory is not homogeneous, meaning that the successive layers of kernel distributions are not concentric. Some areas in the periphery are more used than others. Thus, the kernel values are poorly correlated with the degree of intrusion [42]. Including the past usage of the location in the NPI resides in the fact that the level of threat from neighbours may be higher in areas preferentially used by the resident chimpanzees, as it may impair future usage of this area, and that residents may then be more inclined to respond strongly in these areas of importance, as seen in various primate species [60,61] and as predicted by theoretical models [44,62]. 'F' represents the frequency of occurrence of IGE, based on the number of observation days between consecutive encounters. Over a time period of interest $j$ (yearly on the territory models, monthly on the NPI models), the average ($\mu$) value of the product of the two spatial measures is multiplied by 'F' as the average frequency of IGE occurrence. The three components vary in the same direction, 'I' being larger when the encounter is closer to the centre, 'K' being larger when the encounter takes place in a heavily used area, 'F' being larger when the frequency of occurrence is high. A large NPI value reflects a stronger neighbour pressure. The NPI considers all the intrusions and interactions from all neighbouring groups, even from those which are not habituated to human

observers, and from which information regarding their relative power is not available. Instead of reflecting the power of neighbouring groups, it reflects how dangerous their intrusions are.

## 2.8. Food availability index

We used a food availability index (FAI) established in previous studies in this population [49,50]. This index is calculated on a monthly basis and combines chimpanzee fruiting phenology scores (absence/ presence of mature fruits), density of tree species and mean basal areas of each tree species. Phenology trails were established on each group's territory; densities and basal areas were measured separately for each group (except for East group for which density measures were based on the mean of the densities in the other groups). Thus, the index reflects local variations and differences in food productivity across each group's territories. Depending on the time-scale of analyses, we either used the monthly values, or averaged these monthly values for each year in yearly analyses.

## 2.9. Statistical analysis

We fitted six linear mixed models (LMM [63]) with Gaussian error structure, in a three by three design, using the function *lmer* of the R-package 'lme4' (v. 1.1-14) [64] to analyse how yearly territory sizes ($N = 54$, models 1A, 1B and 1C) and monthly perceived neighbour pressure (only for months in which IGE occurred, $N = 202$, models 2A, 2B and 2C) are influenced by demographic and environmental variables. Results from supplementary models on neighbour pressure that included all months even if no IGE occurred did not differ from models 2A, 2B and 2C (electronic supplementary material, tables S6–S8). Models were fitted using R v. 3.4.0 (R Core Team 2017). Demographic covariates (number of males, number of mature individuals and group size) were modelled using a within-group centring approach [46], teasing apart the within and between group effects of the number of males, number of mature individuals and group size. In this approach, the deviation from the overall group mean defines the within-group effect of the demographic variables. Within-group effects depict how much the variation of the demographic variables within a group influence variation in the response variable (i.e. how much an increase of the number of males within a group is associated with a larger or smaller territory and/or more or less perceived neighbour pressure). Between-group demographic effects reflect differences between the mean of each group. The relationship between predictor variables and dependent variables may vary and differ depending on the considered level of aggregation (within-group and between-group), thus this approach enables the analysis of different levels of aggregation within the same model [46] and avoids a sample size uniquely corresponding to the number of groups. Variance inflation factors [65], derived using the function *vif* from the R-package 'car' [66] and applied to a standard linear model excluding the random effects and slopes, revealed high multi-collinearity between the mean number of males, the mean number of mature individuals and the mean group size per group (maximum variance inflation factor (VIF): 194 and 206 in territory and neighbour pressure models, respectively, when the three demographic variables are included in the same model). Thus, in the two analyses, we fitted separate LMM, one with the within-group centred number of weaned individuals (proxy for group size, models 1A and 2A), one with the within-group centred number of males (models 1B and 2B) and one with the within-group centred number of mature individuals (models 1C and 2C). We used the AIC for model comparison [67] to define which of the models best explains the variance in the response. Model comparisons were carried out using the R-package 'AICcmodavg' (v. 2.2-1) with the function *aictab*. In both analyses, availability of ripe fruits (food availability) was included as test predictor (yearly mean of monthly values for territory size models, monthly values for the neighbour pressure models). The full tumescent swelling ratio and the proportion of nulliparous females were additional test predictors in the neighbour pressure models. We controlled for observation time in all models and, in the models on neighbour pressure, we added the sine and cosine of the month numbers as control predictors to account for potential monthly seasonal variation [68]. All continuous predictors were z-transformed to a mean of zero and a standard deviation of one.

To keep type I error rates at the nominal level of 5%, we included in each model the random slopes of all test predictors within the random effect of 'group identity', but not the correlation parameters between random intercepts and random slope terms, as in each model these correlations were not identifiable [69,70]. The assumptions of normally distributed and homogeneous residuals were assessed by visually inspecting a qqplot and a plot of residuals against fitted values. None of these plots indicated obvious deviation from these assumptions. We checked for model stability by

**Table 1.** Model comparisons between models using group size, models using the number of adult males and models using the number of mature individuals as measures of a group's competitive ability.

| response | predictor of competitive ability (model reference) | d.f. | logLik | AIC | delta AIC | AIC weight |
|---|---|---|---|---|---|---|
| territory | group size (model 1A) | 10 | −144.185 | 308.4 | 0.00 | 0.939 |
| territory | number of adult males (model 1B) | 10 | −147.477 | 315.0 | 6.58 | 0.035 |
| territory | number of mature individuals (model 1C) | 10 | −147.760 | 315.5 | 7.15 | 0.026 |
| neighbour pressure | group size (model 2A) | 18 | −328.167 | 692.3 | 0.00 | 0.986 |
| neighbour pressure | number of adult males (model 2B) | 18 | −335.519 | 707.0 | 14.7 | 0.001 |
| neighbour pressure | number of mature individuals (model 2C) | 18 | −332.433 | 700.9 | 8.53 | 0.014 |

**Table 2.** Effect of group size on territory size (model 1A). Territory sizes are the annual 95% fixed-kernel of the total ranging for each community. Estimated variance components for the random effects and residuals come from the full model (model 1A). The column 'terms' specifies whether the row refers to a random intercept or random slope component. Marginal effect sizes ($R^2$), counting for the variance explained by fixed effects, was 0.68, while conditional $R^2$, counting for the variance of both fixed and random effects, was 0.84. The p-values in italics indicate a statistically significant effect ($p < 0.05$).

| random effect | terms | variance | estimate | s.e. | $\chi^2$ | 95% CI | d.f. | p |
|---|---|---|---|---|---|---|---|---|
| group | intercept | 8.224 | 4.091 | 3.747 | | −3.293; 11.666 | | |
| group | within group number of individuals[a] | 0.066 | −0.210 | 0.187 | 0.581 | −0.582; 0.183 | 1 | 0.445 |
| | between groups number of individuals[a] | n.a. | 0.781 | 0.178 | 7.223 | 0.436; 1.130 | 1 | *0.007* |
| group | food availability[a,c] | 0.000 | −1.085 | 0.521 | 3.726 | −2.166; −0.035 | 1 | 0.053 |
| group | observation hours[b,d] | 0.939 | 1.638 | 0.716 | 3.746 | 0.248; 3.097 | 1 | 0.052 |
| residual | | 10.097 | | | | | | |

[a]Test predictor.
[b]Control predictor.
[c]z-Transformed, mean and s.d. of the original values were 1.76 and 0.88, respectively.
[d]z-Transformed, mean and s.d. of the original values were 2267.52 and 973.63, respectively, before being log-transformed.

excluding each level of the random effect one at a time from the data and comparing the model estimates derived from these subsets of the data with those derived from the full dataset. This revealed that our models are stable. We used likelihood ratio test (R function *anova* with argument test set to 'Chisq' [71]) to establish the significance of the full as compared to the null models (comprising only control predictors, random effects and slopes). To allow for a likelihood ratio test, we fitted the models using maximum likelihood rather than restricted maximum likelihood [71]. The p-values for the individual effects were based on likelihood ratio tests comparing the full with respective reduced models (R function *drop1*). Statistics inherent to model comparisons are provided in table 1. Results of models 1A and 2A are provided in tables 2 and 3, respectively, while results from models 1B, 1C, 2B and 2C are found in electronic supplementary material, tables S2, S3, S4 and S5, respectively.

**Table 3.** Effect of group size on monthly perceived neighbour pressure (model 2A). Determinants of monthly neighbour pressure perceived for a model using group size (model 2A). Estimated variance components for the random effects and residuals come from the full model. The column 'terms' specifies whether the row refers to a random intercept or random slope component. Marginal effect sizes ($R^2$), counting for the variance explained by fixed effects, was 0.22, while conditional $R^2$, counting for the variance of both fixed and random effects, was 0.30. The $p$-values in italics indicate a statistically significant effect ($p < 0.05$).

| random effect | terms | variance | estimate | s.e. | $\chi^2$ | 95% CI | d.f. | p |
|---|---|---|---|---|---|---|---|---|
| group | intercept | 0.000 | −1.450 | 0.256 | | −1.959; −0.959 | | |
| group | within group number of individuals[a] | 0.004 | 0.060 | 0.042 | 1.688 | −0.024; 0.142 | 1 | 0.193 |
| | between groups number of individuals[a] | n.a. | −0.069 | 0.011 | 9.283 | −0.092; −0.046 | 1 | *0.002* |
| group | food availability[a,c] | 0.000 | 0.069 | 0.113 | 0.342 | −0.143; 0.316 | 1 | 0.558 |
| group | full tumescent swelling ratio[a,d] | 0.034 | −0.025 | 0.133 | 0.032 | −0.295; 0.231 | 1 | 0.856 |
| group | proportion of nulliparous females[a,e] | 0.000 | 0.017 | 0.096 | 0.032 | −0.162; 0.226 | 1 | 0.857 |
| group | observation hours[b,f] | 0.053 | 0.016 | 0.151 | 0.009 | −0.297; 0.306 | 1 | 0.920 |
| group | sin (month)[b,g] | 0.000 | 0.419 | 0.120 | 7.423 | 0.193; 0.666 | 1 | *0.006* |
| group | cos (month)[b,g] | 0.000 | 0.207 | 0.158 | 1.546 | −0.118; 0.528 | 1 | 0.213 |
| residual | | 1.425 | | | | | | |

[a]Test predictor.
[b]Control predictor.
[c]z-Transformed, mean and s.d. of the original values were 1.79 and 1.35, respectively.
[d]z-Transformed, mean and s.d. of the original values were 0.085 and 0.070, respectively.
[e]z-Transformed, mean and s.d. of the original values were 0.081 and 0.061, respectively.
[f]z-Transformed, mean and s.d. of the original values were 241.52 and 82.33, respectively, before being log-transformed.
[g]Transformed into a circular radiant variable.

# 3. Results

## 3.1. Socio-ecological determinants of territory size and neighbour pressure

Yearly territory sizes varied both between and within groups (range across all groups: 6.42–36.59 km$^2$; electronic supplementary material, table S1 and figure S1). Full-null model comparisons for model 1A (group size), model 1B (number of males) and model 1C (number of mature individuals) showed that territory size was significantly affected by the test predictors (model 1A: likelihood ratio test (LRT): $\chi^2 = 11.36$, d.f. = 3, $p = 0.009$, table 2; model 1B: LRT: $\chi^2 = 12.15$, d.f. = 3, $p = 0.006$, electronic supplementary material, table S2; model 1C: LRT: $\chi^2 = 11.82$, d.f. = 3, $p = 0.007$, electronic supplementary material, table S3). The test predictors, when using group size as a proxy for dominance, significantly affected neighbour pressure (model 2A: LRT: $\chi^2 = 11.70$, d.f. = 5, $p = 0.039$, table 3), but when including number of males, the test predictors did not affect neighbour pressure (model 2B: LRT: $\chi^2 = 4.45$, d.f. = 5, $p = 0.48$, electronic supplementary material, table S4). The test predictors also did not significantly affect

neighbour pressure when including the number of mature individuals (model 2C: LRT: $\chi^2 = 9.85$, d.f. = 5, $p = 0.08$, electronic supplementary material, table S5). Model comparisons based on the AIC revealed that the most parsimonious models for both response variables were those including group size (models 1A, 2A, table 1), rather than the number of males (models 1B, 2B, table 1) or the number of mature individuals (models 1C, 2C, table 1).

Following demographic differences between the groups, territory size increased with group size (model 1A: estimate ± s.e. = $0.78 \pm 0.17$, d.f. = 1, $p = 0.007$; table 2), with the number of males (model 1B: estimate ± s.e. = $4.76 \pm 1.35$, d.f. = 1, $p = 0.016$; electronic supplementary material, table S2) and with the number of mature individuals (model 1C: estimate ± s.e. = $0.96 \pm 0.22$, d.f. = 1, $p = 0.008$; electronic supplementary material, table S3). In addition, territory sizes tended to decrease with increasing food availability (table 2). The second analysis showed that perceived neighbour pressure is lower for larger groups (model 2A: estimate ± s.e. = $-0.07 \pm 0.01$, d.f. = 1, $p = 0.002$; table 3; electronic supplementary material, table S6), but neither female attractiveness nor food availability significantly impacted perceived neighbour pressure (table 3; electronic supplementary material, table S6). However, we detected a seasonal effect on perceived neighbour pressure (model 2A, comparison of the full model with a reduced model lacking the sine and cosine of the months—LRT: $\chi^2 = 9.19$, d.f. = 2, $p = 0.01$), with lower perceived neighbour pressure during the longer rainy season from June to October (table 3; electronic supplementary material, table S6).

## 3.2. Within-group versus between-group effects

Demographic between-group effects are better predictors of territory size (table 2; electronic supplementary material, tables S2 and S3) and perceived neighbour pressure (table 3; electronic supplementary material, tables S4 and S5) than demographic within-group effects. The between-group perspective showed that larger groups had larger territories than smaller groups (model 1A; figure 1a; table 2; estimate ± s.e. = $0.78 \pm 0.17$, d.f. = 1, $p = 0.007$), groups with more males had larger territories compared to those with fewer males (model 1B; figure 1b; electronic supplementary material, table S2; estimate ± s.e. = $4.76 \pm 1.35$, d.f. = 1, $p = 0.016$) and groups with more mature individuals had larger territories than those with less mature individuals (model 1C; figure 1c; electronic supplementary material, table S3; estimate ± s.e. = $0.96 \pm 0.22$, d.f. = 1, $p = 0.008$). In addition, larger groups faced lower neighbour pressure than smaller groups (model 2A; figure 2a; table 3; estimate ± s.e. = $-0.07 \pm 0.01$, d.f. = 1, $p = 0.002$). This was also reflected by lower perceived neighbour pressure for groups with more mature individuals (model 2C; figure 2b; electronic supplementary material table S5; estimate ± s.e. = $-0.085 \pm 0.02$, d.f. = 1, $p = 0.006$).

From a within-group perspective, the only nearly significant within-group effect was the positive relation between number of males and territory size (model 1B; figure 1d; electronic supplementary material, table S2; estimate ± s.e. = $1.10 \pm 0.48$, d.f. = 1, $p = 0.052$), suggesting that increasing number of males within the group predicts an increase of the group's territory. We found neither within-group effects of group size on territory sizes (model 1A: estimate ± s.e. = $-0.21 \pm 0.18$, d.f. = 1, $p = 0.445$; table 2), or on perceived neighbour pressure (model 2A: estimate ± s.e. = $0.06 \pm 0.04$, d.f. = 1, $p = 0.193$; table 3). Similarly, we did not find within-group effects of the number of mature individuals on territory sizes (model 1C: estimate ± s.e. = $-0.31 \pm 0.18$, d.f. = 1, $p = 0.144$; electronic supplementary material, table S3), or on perceived neighbour pressure (model 2C: estimate ± s.e. = $0.05 \pm 0.04$, d.f. = 1, $p = 0.202$; electronic supplementary material, table S5).

# 4. Discussion

Our study uses long-term data on four neighbouring groups of chimpanzees to investigate demographic effects of the number of males, adults or community size, on components of BGC, such as territory size and perceived neighbour pressure. Our findings support the notion of the inter-group dominance hypothesis, as larger groups hold larger territories and experience lower neighbour pressure. In this population, group size rather than number of adult males reflects the competitive ability of a group. Long-term data within and between several groups has helped clarify the relation between group size and territory size in this population [23,28].

Our study confirms that the number of adult males predicts territory size, as group territories increased with increasing numbers of males, and groups with more males had larger territories than groups with fewer males. One reason may be that more males increase male party sizes leading to better territorial

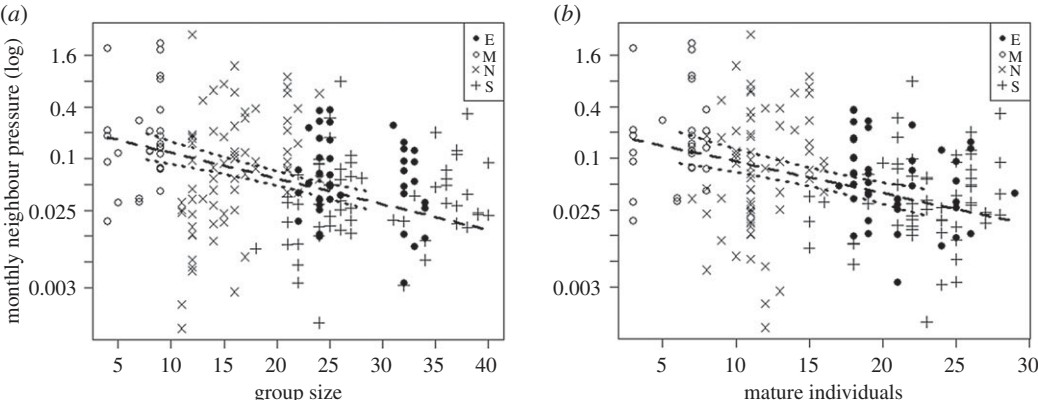

**Figure 1.** Effects of demographic variables on yearly territory sizes; (*a*) between-group effect of group size; (*b*) between-group effect of the number of adult males; (*c*) between-group effect of the number of mature individuals (adult and adolescent males and females); (*d*) within-group effect of the number of mature males. The dashed lines show the fitted model and the dotted lines the 95% confidence intervals. Letters and associated symbols correspond to the different communities: East group (E), Middle group (M), North group (N) and South group (S).

**Figure 2.** Demographic between-group effects on the perceived neighbour pressure. (*a*) Between-group effect of group size; (*b*) between-group effect of the number of mature individuals (adult and adolescent males and females). The dashed line shows the fitted model and the dotted lines the 95% confidence interval. Letters and associated symbols correspond to the different communities: East group (E), Middle group (M), North group (N) and South group (S).

maintenance due to more effective border patrols. Such a relation is suggested by the association between patrolling behaviour and number of males found in previous studies [57,72]. The number of males is correlated with inter-group killings [21], and inter-group killings are known to lead to territorial expansion [25]. The increase of territory size with an increasing number of males in Taï chimpanzees, therefore, is consistent with the dominant role of males in group dominance. The model comparison, however, suggests that, when comparing the groups, all individuals appear important in territory acquisition and defence. This is due to the fact that larger groups (including independent offspring) hold larger territories and perceive lower neighbour pressure. The main effect of larger groups holding larger territories might be explained by energetic constraints: while increases in group size increase within-group feeding competition, the benefits of reducing feeding competition due to an expanded territory outweigh the costs of risky between-group aggression. Such a conclusion was already suggested by previous findings in this and other populations, since larger party sizes were associated with longer travelling distances [73–75]. It is also consistent with findings that larger groups occupy larger territories in a variety of social species [8–16].

A possible reason for group size acting as the main mechanism of group dominance in this population could reside in the bisexually-bonded social system of the Taï chimpanzees [20,30,33] where males and females use the same community home-range, rather than females living in segregated core areas as in male-bonded populations [27,32]. In bisexually-bonded systems, both sexes participate in the majority of territorial border patrols [35] and can be involved in violent interactions with neighbours [20,76]. The grouping patterns in this social system may explain why group size better predicts variation in territory size and perceived neighbour pressure than number of males. Regular female participation in territorial behaviour and inter-group aggression [20,35] has the potential to increase party sizes, tilting any imbalance of power [24] in favour of their own group. On the other hand, the formation of larger parties from both sides, due to female participation, may also flatten the odds of imbalance of power, resulting in more balanced conflicts [20,35]. For example, in bonobos, where females are co-dominant with males and mixed-sex parties are frequent [77], reports of between-group violence are relatively rare [78], potentially as the odds of imbalance of power are less likely to occur. Following this reasoning and other studies in the Taï population [20,35], it is possible that the bisexually-bonded social system of Taï chimpanzees favours a lower inter-group aggression frequency and lower occurrence of inter-group killings than in male-bonded populations [21]. The model comparison shows that group size differences between groups explains better the perceived neighbour pressure than the number of mature individuals. This difference may come from a more frequent participation of individuals in border patrols and inter-group aggression when the number of offspring is high. Ngogo males who have more offspring in the group show this tendency [57], as well as Taï females who have juvenile sons in the group [79]. Overall, our findings suggest that the main mechanism of group dominance in Taï chimpanzees is acting through group size, which is in line with the imbalance of power hypothesis: larger groups lead to larger parties which confer a competitive advantage toward neighbours during inter-group conflicts.

Even if demographic effects are important in chimpanzee BGC, this competition seems also modulated by food availability and distribution. First, when food availability is high, annual territories tend to be smaller, indicating that ranging patterns are adjusted according to the energetic needs (also seen in the dynamics of fission–fusion [74]). Second, higher neighbour pressure occurs at the end of the dry season, when food availability decreases. This may be due to an increase of the usage of border areas during this period. Border areas of the territories being generally underused [48,80,81], they potentially contain a higher density of food resources leading to higher rates of inter-group encounters in these locations [82]. Spatial distribution of food resources influence grouping patterns in chimpanzees [82–84], thus further studies investigating the ranging and grouping behaviour in relation to local distribution of food resources and neighbour pressure could help understand the proximate impact of resources on the intensity of BGC.

The bisexual grouping pattern of Taï chimpanzees has supposedly emerged due to a high predation level and relatively low female feeding competition due to high food abundance, as compared to other chimpanzee populations [30]. Other, possibly associated, factors differing between Taï and these populations can also play a role, such as the high risk of infanticide by in-group males (Mahale [85], Kanyawara [86], Kasekela [87]), by out-group males (Sonso [88], Mahale [89], Ngogo [90,91], Kasekela [92]) and by out-group females (Sonso [76]), that may prevent females from participating in territorial defence with males in eastern populations. In Taï, low in-group infanticide risk [23], coupled with bisexual grouping patterns may have facilitated bonding opportunities between sexes [93]. This, in turn coupled with low between-group infanticide may lead to increased likelihood of participation of females in territorial defence.

Differing participation of the sexes in territorial defence in social species depends on a variety of factors, such as the dominance structure [94], dispersal and breeding system [95], defence strategies (mate defence versus resource defence) [96] and fitness outcomes individuals of different sexes may benefit from. In non-pair breeding species with female philopatry, participation of both sexes to territorial defence is common, though easily explained by different adaptive strategies for each sex: males defend access to females, while females defend their resources. This is seen in a variety of primate species (reviewed in [97]), in spotted hyenas [98] and in African lions [99], where both sexes gain mutual fitness benefits from these differing strategies. Taï chimpanzees, like other chimpanzee populations, present a male philopatric system, in which territorial aggression from females may not be expected [100]. However, Taï females may gain fitness benefits by participating in territorial defence alongside males. This strategy is probably facilitated by tight bonds among individuals [93]. Both sexes joining territorial patrols lead to larger travelling parties which are in turn more likely to lead to neighbour repulsion and increases in territory size, resulting in benefits for both sexes. Indeed, Taï female's reproductive success is reduced by high levels of neighbour pressure [42], since inter-birth intervals are longer and offspring survival reduced in times of high neighbour pressure. Thus, reducing neighbour pressure through cooperation among males and females may improve female fitness in this population. Males probably also gain fitness benefits via a better survival of their offspring and faster rates of female reproduction. Together with reinforced cohesion due to predation pressure, both sexes benefit from collectively participating in territorial defence. Even if defence strategies may differ between sexes, the mutual fitness benefits inferred in this population may help understanding the conditions in which mechanisms of co-defence by both sexes have emerged in other social species. For example, in a co-dominant primate species, male–female interactions during inter-group conflicts through mutual support [101] and affiliative interactions [102] enable a reduction of the costs linked to these conflicts. More research focus on the association between male–female relationships, participation of both sexes to territorial defence and social incentives [103] will help clarify the socio-ecological determinants of collective territorial defence.

Overall, our results confirm that group dominance acting through group size is a mechanism regulating BGC in the bisexually-bonded Taï population. Larger group size may favour the imbalance of power during conflicts, improve the efficiency of border patrols and enable efficient repulsion of hostile neighbours, leading to mutual benefits for both sexes. Since BGC may act as a strong selective pressure leading to adaptations such as in-group cooperation with non-kin [4,5,35,36], one should expect that similar adaptations in both male and female chimpanzees would have been selected. The relative importance of other selective pressures, such as infanticide risk, population density, resource distribution and male–male competition may then account for most of the differences in association patterns and cooperation found between the various chimpanzee populations.

In early hominins, evidence points toward a male philopatric system [104,105] that would involve a stronger propensity for between-group male aggression [100], but nothing is known about the involvement of females in territorial defence. Reports in hunter-gatherer societies that engage in territorial aggression show that mostly males participate to territorial defence, while females seem to play the role of peacemakers [106]. However, both sexes equally cooperate with non-kin. A possibility for an evolutionary path is that both sexes in the common ancestor between chimpanzees/bonobos on one side and early humans on the other had already evolved strong capacities for cooperation with non-kin under the pressure of BGC, but that the increase in human ancestors' reproductive rates via reduced inter-birth intervals [107], combined with central place settlements [108], increased the division of labour between sexes and limited female capacity to participate in territorial defence. Considering variation in social systems and participation of different sexes in territorial behaviour, across species, and across populations within species, both in human and non-human societies, will offer insights on socio-ecological determinants of territoriality and the role of BGC in shaping cooperation with non-kin.

Ethics. All methods used in this study were non-invasive and were approved by the Ministère de l'Enseignement Supérieur et de la Recherche Scientifique and the Ministère de Eaux et Forêts in Côte d'Ivoire, and the Office Ivoirien des Parcs et Réserves. All aspects of the study comply with the ethics policy of both the Max Planck Society, the Department of Primatology and the Department of Human Ecology, Behavior and Culture of the Max Planck Institute for Evolutionary Anthropology, Germany.
Data accessibility. The datasets supporting this article have been uploaded as part of the electronic supplementary material.
Authors' contributions. S.L. and R.M.W. conceived and designed the study; S.L., C.C. and R.M.W. developed the methods; S.L. carried out the statistical analysis and drafted the manuscript; S.L., C.B., L.S., A.P., C.C. and R.M.W. provided the

data and critically revised and edited the manuscript; C.C. and R.M.W. supervised the project. All authors gave final approval for publication and agree to be held accountable for the work performed therein.

Competing interests. The authors declare no competing interests.

Funding. S.L., C.B., A.P., L.S., C.C. and R.M.W. were supported by the Max Planck Society. Research at the Taï Chimpanzee Project has been funded by the Max Planck Society since 1997. C.C. was supported by the European Research Council (ERC) under the European Union's Horizon 2020 research and innovation programme (grant agreement no. 679787).

Acknowledgements. We thank the Ministère de l'Enseignement Supérieur et de la Recherche Scientifique of Côte d'Ivoire, the Ministère de Eaux et Forêts of Côte d'Ivoire and the Office Ivorien des Parcs et Réserves for permissions to conduct the study. We are grateful to the Centre Suisse de Recherches Scientifiques en Côte d'Ivoire and the staff members of the Taï Chimpanzee Project for their long-term support. The Max Planck Society provides core funding for the Taï Chimpanzee Project since 1997.

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
