## [Reviewer comments · Royal Society Open Science]

Review History

Decision letter (RSOS-200577.R0)

Dear Mr Lemoine

On behalf of the Editors, I am pleased to inform you that your Manuscript RSOS-200577 entitled "Group dominance increases territory size and reduces neighbour pressure in wild chimpanzees" has been accepted for publication in Royal Society Open Science subject to minor revision in accordance with the referee suggestions. Please find the referees' comments at the end of this email.

The reviewers and handling editors have recommended publication, but also suggest some minor revisions to your manuscript. Therefore, I invite you to respond to the comments and revise your manuscript.

- Ethics statement

If your study uses humans or animals please include details of the ethical approval received, including the name of the committee that granted approval. For human studies please also detail

whether informed consent was obtained. For field studies on animals please include details of all permissions, licences and/or approvals granted to carry out the fieldwork.

- Data accessibility

If you wish to submit your supporting data or code to Dryad (<http://datadryad.org/>), or modify your current submission to dryad, please use the following link:
<http://datadryad.org/submit?journalID=RSOS&manu=RSOS-200577>

- Competing interests

- Authors' contributions

- Acknowledgements

- Funding statement

Because the schedule for publication is very tight, it is a condition of publication that you submit the revised version of your manuscript before 26-Apr-2020. Please note that the revision deadline will expire at 00.00am on this date. If you do not think you will be able to meet this date please let me know immediately.

If your manuscript is newly submitted and subsequently accepted for publication, you will be asked to pay the article processing charge, unless you request a waiver and this is approved by Royal Society Publishing. You can find out more about the charges at <https://royalsocietypublishing.org/rsos/charges>. Should you have any queries, please contact opscience@royalsociety.org.

Once again, thank you for submitting your manuscript to Royal Society Open Science and I look

forward to receiving your revision. If you have any questions at all, please do not hesitate to get in touch.

Kind regards,

on behalf of Dr Alecia Carter (Associate Editor) and Kevin Padian (Subject Editor)
 openscience@royalsociety.org

Associate Editor Comments to Author (Dr Alecia Carter):

Dear authors,

My apologies for taking some time to make this decision--as I'm sure you understand, lockdown with a toddler gives one very little time for work.

I enjoyed reading your manuscript and I am satisfied with your responses that you provided to the original reviews. I found several typos that should be incorporated when possible, and one very minor point that may require clarification:

L26: to -> in

L41: into -> in

L42: population scale -> population-scale

L51: preys -> prey

LL49-50: is higher reproductive output an indication of group dominance? Wild dogs are cooperative breeders, so larger groups could have higher RS because they have more helpers.

L63: case -> cases

L64: are -> is

L68 & 75: Pan. -> P. (or Pan, without the period, as it is not abbreviated)

L69: suggested also -> also suggested

L81: remove "of"

L88: neighbour's -> neighbours'

LL98-102 and 104-107: these sentences are not complete. Perhaps add "measured using" before the colon (L100) and "As a measure of" before "Dominance being determined" (L104).

L125: within and between group -> within- and between-group

L131: use abbreviated Latin name

L135: enables the capture -> captures

L144: insert space between numeral and unit

LL153-155 and elsewhere: are these count data approximately normally distributed? If not, present the median \pm IQR as this is a more appropriate measure of central tendency for Poisson-distributed data.

L170: age at -> an age at

L190: how -> of

LL222-223: in both instances, change "have" to "has" OR "encounter" to "encounters"

L226: invert -> inverse

L240: enables to consider -> considers (or: enables the consideration of)

L242: are -> is

LL268-269: "This approach enables to not generalize within-group effects from between-group effects" This phrase doesn't make sense, but I'm not sure what the authors intend. Please rephrase.

LL307-308: make "2" superscript

LL337 & 365: less -> fewer
LL376, 422: between group -> between-group
L383: system -> systems (or insert "a" after "In")
L409: to a higher -> to higher
L423: to -> in

Author's Response to Decision Letter for (RSOS-200577.R0)

See Appendix A.

Decision letter (RSOS-200577.R1)

Dear Mr Lemoine,

It is a pleasure to accept your manuscript entitled "Group dominance increases territory size and reduces neighbour pressure in wild chimpanzees" in its current form for publication in Royal Society Open Science. The comments of the reviewer(s) who reviewed your manuscript are included at the foot of this letter.

on behalf of Dr Alecia Carter (Associate Editor) and Kevin Padian (Subject Editor)
openscience@royalsociety.org

Follow Royal Society Publishing on Twitter: [@RSocPublishing](https://twitter.com/RSocPublishing)
Follow Royal Society Publishing on Facebook:
<https://www.facebook.com/RoyalSocietyPublishing.FanPage/>
Read Royal Society Publishing's blog: <https://blogs.royalsociety.org/publishing/>

Appendix A

Max Planck Institute for Evolutionary Anthropology

Max-Planck-Institut für evolutionäre Anthropologie

MPI für evolutionäre Anthropologie • Deutscher Platz Nr. 6 • 04103 Leipzig • Germany

MAX-PLANCK-GESellschaft

**Department of Primatology,
Department of Human Behavior, Ecology and Culture**
Sylvain Lemoine
Tel: +33 (0)652-4957-97
sylvain_lemoine@eva.mpg.de

Editorial Office of *Royal Society Open Science*

Leipzig, 21th April 2020

Dear Editors,

We are very pleased that our paper entitled “*Group dominance increases territory size and reduces neighbour pressure in wild chimpanzees*” is now accepted for publication in Royal Society Open Science. We have addressed the minor revisions suggested by the Associate Editor Dr. Alecia Carter.

You will find, below this letter, the comments and suggested changes in which we have included our responses (in *italic*). We have also included below the original manuscript previously submitted including the track changes, so that you can assess where and how the changes were incorporated. The revised manuscript without track changes is submitted separately.

Sincerely,

Sylvain Lemoine, Ph.D. Candidate

Editors' comments from submission to *Royal Society Open Science* (ref: RSOS-200577). Responses from the authors are provided in *italic*. Line numbers refer to the revised manuscript without track changes, submitted as a separate document.

Dear Mr Lemoine

On behalf of the Editors, I am pleased to inform you that your Manuscript RSOS-200577 entitled "Group dominance increases territory size and reduces neighbour pressure in wild chimpanzees" has been accepted for publication in Royal Society Open Science subject to minor revision in accordance with the referee suggestions. Please find the referees' comments at the end of this email.

The reviewers and handling editors have recommended publication, but also suggest some minor revisions to your manuscript. Therefore, I invite you to respond to the comments and revise your manuscript.

(...)

on behalf of Dr Alecia Carter (Associate Editor) and Kevin Padian (Subject Editor)
openscience@royalsociety.org

Dear Editors,

We are grateful that the revisions to our paper after initial submission to Proceedings B have been found satisfying, and we think that these changes have helped improving the soundness and readability of our findings.

Hereafter we have included the responses of the editor's comments in "italic". The line referring corresponds to the newest version of the manuscript. At the end of this document, we present a copy of the earlier version of the manuscript, in which all changes and modifications appear with track changes.

Associate Editor Comments to Author (Dr Alecia Carter):

Dear authors,

My apologies for taking some time to make this decision--as I'm sure you understand, lockdown with a toddler gives one very little time for work.

I enjoyed reading your manuscript and I am satisfied with your responses that you provided to the original reviews. I found several typos that should be incorporated when possible, and one very minor point that may require clarification:

L26: to -> in *This was changed line 26*

L41: into -> in *This was changed line 41*

L42: population scale -> population-scale *This was changed line 42*

L51: preys -> prey *This was changed line 51*

LL49-50: is higher reproductive output an indication of group dominance? Wild dogs are cooperative breeders, so larger groups could have higher RS because they have more helpers.

Thank you for this comment. We have replaced this reference by a more relevant example in spotted hyenas (lines 49-50).

L63: case -> cases *This was changed line 64*

L64: are -> is *This was changed line 65*

L68 & 75: Pan. -> P. (or Pan, without the period, as it is not abbreviated) *This was changed lines 69 and 76*

L69: suggested also -> also suggested *This was changed line 70*

L81: remove "of" *This was changed line 82*

L88: neighbour's -> neighbours' *This was changed line 89*

LL98-102 and 104-107: these sentences are not complete. Perhaps add "measured using" before the colon (L100) and "As a measure of" before "Dominance being determined" (L104).
Changes were made line 101, and lines 105 to 108 ("As potential proxies for group dominance, determined by the competitive ability of a group, we used the number of adult individuals of the dominant sex of the group (males), the number of adults and adolescent males and females (number of mature individuals) and group size (number of independent individuals).")

L125: within and between group -> within- and between-group *This was changed line 126*

L131: use abbreviated Latin name *Change was made line 132*

L135: enables the capture -> captures *This was modified line 136*

L144: insert space between numeral and unit *This was changed line 145*

LL153-155 and elsewhere: are these count data approximately normally distributed? If not, present the median \pm IQR as this is a more appropriate measure of central tendency for Poisson-distributed data.
Number of follows and hours of observation are indeed count data, so we have followed the editor's suggestion and have provided the median and IQR instead of mean and SD (lines 154 to 156). Regarding the provision of mean and sd of original values of variables in Table 2 and Table 3 (as well in supplementary tables), we have left these information as these variables have been z-transformed for analysis. Providing mean and SD values for z-transformed variables allows to retrieve the values of original data.

L170: age at -> an age at *This was changed line 171*

L190: how -> of *This was changed line 191*

LL222-223: in both instances, change "have" to "has" OR "encounter" to "encounters"
We have made these changes line 223-224

L226: invert -> inverse *This was changed line 227*

L240: enables to consider -> considers (or: enables the consideration of) *This was changed line 241*

L242: are -> is *This was changed line 243*

LL268-269: "This approach enables to not generalize within-group effects from between-group effects"
This phrase doesn't make sense, but I'm not sure what the authors intend. Please rephrase.
We have changed this statement by another sentence that makes our point more understandable (lines 269 to 272) – ("The relationship between predictor variables and dependent variables may vary and differ depending on the considered level of aggregation (within-group and between-group), thus this approach enables the analysis of different levels of aggregation within the same model [46] and avoids a sample size uniquely corresponding to the number of groups").

LL307-308: make "2" superscript *We have made these changes lines 310-311*

LL337 & 365: less -> fewer *Changes were made line 340 and 368*

LL376, 422: is between group -> between-group *These changes were made lines 379 and 425*

L383: system -> systems (or insert "a" after "In") *This change was made line 386*

L409: to a higher -> to higher *This change was made line 412*

L423: to -> in *This change was made line 426*

We hope the implemented changes are satisfactory and the MS is acceptable for publication in RSOS. Looking forward to hearing from you.

Sylvain Lemoine

Original manuscript submitted to *Royal Society Open Science* (RSOS-200577), including track changes addressing the editor's comments.

[revised manuscript text omitted]

Our data have been made accessible as Supplementary Information files.

**Author's contributions**

SL and RMW conceived and designed the study; SL, CC and RMW developed the methods; SL
carried out the statistical analysis and drafted the manuscript; SL, CB, LS, AP, CC and RMW provided
the data and critically revised and edited the manuscript; CC and RMW supervised the project. All
authors gave final approval for publication and agree to be held accountable for the work performed
therein.

**Competing interests.**

The authors declare no competing interests.

**Funding**

This work was supported by the Max Planck Society.

**Acknowledgments**

We ~~thank~~ ~~are grateful to~~ the Ministère de l'Enseignement Supérieur et de la Recherche
Scientifique of Côte d'Ivoire, the Ministère de Eaux et Forêts of Côte d'Ivoire and the Office Ivoirien des
Parcs et Réserves for permissions to conduct the ~~research~~ study; ~~We are grateful to also~~ to the Centre
Suisse de Recherches Scientifiques and the staff members of the Taï Chimpanzee Project for their long-
term support. The Max Planck Society provides core funding for the Taï Chimpanzee Project since 1997.

**References**

- 1. Smith JM, Parker GA. 1976 The logic of asymmetric contests. *Animal Behaviour* **24**,
159–175. (doi:10.1016/S0003-3472(76)80110-8)
 - 2. Georgiev AV, Klimczuk ACE, Traficonte DM, Maestripieri D. 2013 When Violence
Pays: A Cost-Benefit Analysis of Aggressive Behavior in Animals and Humans. *Evolutionary Psychology*
**11**, 147470491301100. (doi:10.1177/147470491301100313)
 - 3. Hardy ICW, Briffa M, editors. 2013 *Animal contests*. Cambridge: Cambridge Univ.
Press.
 - 4. Bowles S. 2006 Group Competition, Reproductive Leveling, and the Evolution of Human
Altruism. *Science* **314**, 1569–1572. (doi:10.1126/science.1134829)

- 5. Radford AN, Majolo B, Aureli F. 2016 Within-group behavioural consequences of
between-group conflict: a prospective review: Table 1. *Proceedings of the Royal Society B: Biological*
*Sciences* **283**, 20161567. (doi:10.1098/rspb.2016.1567)
- 6. Clutton-Brock T, Sheldon BC. 2010 Individuals and populations: the role of long-term,
individual-based studies of animals in ecology and evolutionary biology. *Trends in Ecology & Evolution*
**25**, 562–573. (doi:10.1016/j.tree.2010.08.002)
- 7. Crofoot MC, Wrangham RW. 2010 Intergroup Aggression in Primates and Humans: The
Case for a Unified Theory. In *Mind the Gap* (eds PM Kappeler, J Silk), pp. 171–195. Berlin, Heidelberg:
Springer Berlin Heidelberg. (doi:10.1007/978-3-642-02725-3_8)
- 8. Mosser A, Packer C. 2009 Group territoriality and the benefits of sociality in the African
lion, *Panthera leo*. *Animal Behaviour* **78**, 359–370. (doi:10.1016/j.anbehav.2009.04.024)
- 9. Harris T. 2007 Testing mate, resource and infant defence functions of intergroup
aggression in non-human primates: issues and methodology. *Behaviour* **144**, 1521–1535.
(doi:10.1163/156853907782512128)
- 10. Watts HE, Holekamp KE. 2009 Ecological Determinants of Survival and Reproduction in
the Spotted Hyena. *J. Mammal.* **90**, 461–471. (doi:10.1644/08-MAMM-A-136.1)
- ~~10. Gusset M, Macdonald DW. 2010 Group size effects in cooperatively breeding African
wild dogs. *Animal Behaviour* **79**, 425–428. (doi:10.1016/j.anbehav.2009.11.021)~~

[revised manuscript text omitted]

801 *News Rev.* **2**, 78–88.

802

803